# Playing with heart and soul...and genomes: sports implications and applications of personal genomics

Jennifer K. Wagner

Center for the Integration of Genetic Healthcare Technologies, Division of Translational Medicine and Human Genetics, University of Pennsylvania, United States

## ABSTRACT

Whether the integration of genetic/omic technologies in sports contexts will facilitate player success, promote player safety, or spur genetic discrimination depends largely upon the game rules established by those currently designing genomic sports medicine programs. The integration has already begun, but there is not yet a playbook for best practices. Thus far discussions have focused largely on whether the integration would occur and how to prevent the integration from occurring, rather than how it could occur in such a way that maximizes benefits, minimizes risks, and avoids the exacerbation of racial disparities. Previous empirical research has identified members of the personal genomics industry offering sports-related DNA tests, and previous legal research has explored the impact of collective bargaining in professional sports as it relates to the employment protections of the Genetic Information Nondiscrimination Act (GINA). Building upon that research and upon participant observations with specific sports-related DNA tests purchased from four direct-to-consumer companies in 2011 and broader personal genomics (PGx) services, this anthropological, legal, and ethical (ALE) discussion highlights fundamental issues that must be addressed by those developing personal genomic sports medicine programs, either independently or through collaborations with commercial providers. For example, the vulnerability of student-athletes creates a number of issues that require careful, deliberate consideration. More broadly, however, this ALE discussion highlights potential sports-related implications (that ultimately might mitigate or, conversely, exacerbate racial disparities among athletes) of whole exome/genome sequencing conducted by biomedical researchers and clinicians for non-sports purposes. For example, the possibility that exome/genome sequencing of individuals who are considered to be non-patients, asymptomatic, normal, etc. will reveal the presence of variants of unknown significance in any one of the genes associated with hypertrophic cardiomyopathy (HCM), long QT syndrome (LQTS), Marfan's syndrome, and other conditions is not inconsequential, and how this information is reported, interpreted, and used may ultimately prevent the individual from participation in competitive sports. Due to the distribution of genetic diversity that reflects our evolutionary and demographic history (including the discernible effects of restricted gene flow and genetic drift associated with cultural constructs of race) and in recognition of previous policies for "leveling" the playing field in competitive sports based on "natural" athletic abilities, preliminary recommendations are provided to discourage genetic segregation of sports and to develop best

Corresponding author
Jennifer K. Wagner,
jennifer.wagner@uphs.upenn.edu

practice guidelines for genomic sports medicine programs that will facilitate player success, promote player safety, and avoid genetic discrimination within and beyond the program.

## INTRODUCTION

In March 2013 the American College of Medical Genetics and Genomics (ACMG) and the American Academy of Pediatrics (AAP) announced a policy statement on genetic testing and screening of children (*Ross et al., 2013*). As shown in Table 1, the AAP and ACMG jointly and unequivocally opposed a number of practices regarding minors, including school-based genetic screening or testing, routine carrier screening for recessive conditions (such as sickle cell carrier status), and direct-to-consumer testing. Similarly, the AAP and ACMG cautioned against the expansion of newborn screening, warning that such practices may "give rise to 'patients in waiting': individuals with a genetic diagnosis who have no signs or symptoms and may remain asymptomatic for years or decades." The need for genetic counseling was highlighted, specifically in the context of predictive testing (i.e., testing for "the presence of a mutation that will almost certainly give rise to clinical manifestations"), with the AAP and ACMG noting that such counseling is "essential to ensure that parents, guardians, and maturing minors fully understand the limits of genetic knowledge and treatment capabilities as well as the potential for psychological harm, stigmatization, and discrimination." The AAP and ACMG also took positions on access to testing, timing of testing, and access to test results, as shown in Table 2, recognizing parents and children may have differing opinions. In doing so, the organizations emphasized the need to focus on the best medical interests of the child but recognized those interests are "embedded in and dependent on the interests of the family unit."

Also in March 2013, the ACMG announced age-neutral recommendations (*Green et al., 2013*) for reporting of incidental findings in exome and genome sequencing (WES/WGS). Prompted by criticisms and perhaps also confusion (e.g., *Heger, 2013*), the ACMG issued clarifications shortly thereafter (*ACMG, 2013*). Notwithstanding the positions the ACMG affirmed with its joint statement with the AAP, the ACMG recommended the minimum reporting of incidental findings of any variants previously reported as "known" or "expected" to be pathogenic—regardless of the well-documented publication bias of positive results in the academic literature and regardless of the individual's health—in "a set of 57 carefully chosen genes for pathogenic mutations that could indicate the presence of any of 24 disorders where early intervention is likely to reduce or prevent serious morbidity or early mortality" (*Green et al., 2013*). The ACMG reaffirmed its concerns regarding an emerging class of "patients in waiting" who may face significant psychological burdens and exposure to unnecessary surveillance and diagnostic testing

**Table 1  AAP and ACMG positions on genetic testing and screening of youth.**

| | |
|---|---|
| School-based screening or testing | "The AAP and the ACMG do not support school-based genetic screening or testing because the school setting raises concerns about whether the uptake is informed and voluntary, whether privacy and confidentiality are maintained, and whether appropriate genetic counseling is provided before and after testing." (p. 237) |
| Carrier screening | "The AAP and ACMG do not support routine carrier testing or screening for recessive conditions when carrier status has no medical relevance during minority." (p. 236) |
| Direct-to-consumer testing | "The AAP and the ACMG strongly discourage the use of DTC and home-kit genetic testing of children." (p. 241) |

**Table 2  AAP and ACMG positions on youth access to testing and results.**

| | |
|---|---|
| Access to testing | "If an adolescent is not interested in testing, and the clinical benefits of knowing will not be relevant for years to decades, the adolescent's dissent should be final." (p. 238) |
| | "In the case of predictive testing for childhood-onset conditions or conditions for which childhood interventions will ameliorate future harm…parental authority to determine medical treatment supersedes the minor's preferences with regard to liberty and privacy." (p. 238) |
| | "Health-care providers should be cautious about providing such [predictive genetic] testing to minors without the collaboration of their parents." (p. 238) |
| Timing of testing | "Significant deference should be extended to parents regarding the timing of predictive genetic testing for childhood-onset conditions." (p. 238) |
| | "The AAP and the ACMG continue to support the traditional professional recommendation to defer genetic testing for late-onset conditions until adulthood…" (p. 238) |
| Access to results | "The AAP and the ACMG believe that a request for the results of a genetic test by a mature adolescent should be given priority over his or her parents' requests to conceal the information." (p. 238) |

as the result of learning about incidental findings from WES/WGS. While the ACMG clarified that WES/WGS specifically of youth should be performed "only if there are clear clinical indications", the organization noted that this recommendation to report to youth incidental findings of "a severe, actionable, pathogenic mutation" is not contradictory to or inconsistent with the organization's prior recommendation made with the AAP (*ACMG, 2013*).

## Sports implications of genomic research and medicine

As shown in Table 3, among the 57 genes on the ACMG's list for reporting of variants are a number of genes involved with cardiac conditions, including 11 genes reported to cause Hypertrophic Cardiomyopathy (HCM), three genes reported to cause Long QT Syndrome (LQTS), and seven genes reported to cause Marfan Syndrome (MFS) and related disorders.

HCM (OMIM #192600) has an estimated prevalence of 1 in 500. The condition is clinically variable and genetically heterogeneous. At least 18 genes have been reported, including the 11 genes included on the ACMG list for reporting of incidental findings. Roughly 60–70% of cases of HCM are due to mutations in one or more of the 18 genes. Reliably distinguishing pathogenic variants from variants of unknown significance (VUS) and non-pathogenic variants in these genes presents a major challenge for genomic medicine (*Maron, Maron & Semsarian, 2012*). Notably, roughly 50% of adults

**Table 3 Subset of Genes on the ACMG's Minimum List for Reporting.**

| | |
|---|---|
| Hypertrophic Cardiomyopathy | *MYBPC3, MYH7, TNNT2, TNNI3, TPM1, MYL3, ACTC1, PRKAG2* |
| Long QT Syndrome | *KCNQ1, KCNH2, SCN5A* |
| Marfan Syndrome and related disorders | *FBN1, TGFBR1, TGFBR2, SMAD3, ACTA2, MYLK, MYH11* |

diagnosed with HCM have a family history of either HCM or sudden death at a young age (e.g., *GeneDx, 2011a*). HCM is the leading cause of sudden cardiac arrest in young individuals and athletes (e.g., *Maron et al., 2009*).

LQTS (OMIM #192500) has an estimated prevalence of more than 1 in 3000. Approximately 75% of the cases of LQTS are due to known genetic mutations. Like HCM, LQTS is genetically heterogeneous: while three genes are on the ACMG's minimum list for reporting of incidental findings, more than 12 genes have been reported. Moreover, mutations have shown incomplete penetrance, and LQTS has variable expressivity even within families. Some individuals with LQTS show no clinical symptoms; however, for approximately 10–15% of individuals diagnosed with LQTS, the first symptom is sudden death. While individuals with LQTS may experience fainting or heart palpitations with exercise, others may experience such events while at rest or upon auditory stimulation (e.g., *GeneDx, 2011b*).

MFS (OMIM #154700), estimated to have a worldwide prevalence of 1 in 3000–5000, is a connective tissue disorder involving *FBN1*, a gene marked by considerable pleiotropy (e.g., *Pyeritz, 2002*). Expression is highly variable, but clinical features may include tall stature with disproportionately long limbs, joint hypermobility, ectopia lentis, myopia, mitral valve prolapse, aortic dissection, and aortic root dilation (among others). Genetic testing is considered important to identify pre-symptomatic individuals who could benefit from surveillance (e.g., *GeneDx, 2012*).

Suspected and/or confirmed diagnosis of any of these conditions (HCM, LQTS, or MFS and related disorders) has serious implications for an individual's future eligibility for participation in competitive athletics (even when the individual is asymptomatic). Many experts and policymakers are convinced by anecdotal evidence suggesting that restriction of physical activity is an appropriate measure to prevent sudden cardiac death (SCD) among individuals with one of these conditions (see, e.g., *Maron & Zipes, 2005*). Empiric data, however, are contradictory as to whether moderate to rigorous physical activity has beneficial or adverse effects on risk of SCD (e.g., *Deo & Albert, 2012*). Additionally, a broad restriction on athletic participation overlooks the variation of the static and dynamic components required of different sports that may confer differential risks of SCD (*Vaseghi, Ackerman & Mandapati, 2012*). There is a dearth of empiric data demonstrating that individuals with such genetic variants in genes reported for HCM, LQTS, MFS or related conditions who participate in competitive sports have higher risks of SCD than those individuals with the same genetic variants who refrain from participation in competitive sports. Notably, *Hoffman et al. (2012)* have reported that an *FBN1* mutation is not itself

a predictor for SCD, and O'Mahony and colleagues (*2013*) have described the difficulties and limited power of risk stratification algorithms designed to identify those individuals diagnosed with HCM who are at a high risk of SCD.

Preventing sudden cardiac death in athletes is a high priority. As *Angelini et al. (2013)* noted,

> "...[A]t present, it is probably reasonable to assume that 2% to 3% of the general population has cardiovascular conditions...that seem to pose a high risk to competitive sportspersons. Considering that the young population constitutes approximately 28% of the total United States population, or about 90 million, the 2% to 3% would correspond to about 2 million people. If we consider only the estimated 1.5 to 10 million "young athletes" in the U.S. (usually defined as "regular runners") in any given year, 30,000 to 300,000 of them would be expected to carry high-risk cardiovascular conditions (hr-CVCs). Preliminary estimates suggest that 0.1 to 0.6 per 100,000 young people die suddenly of cardiac causes each year, whereas 2 to 7 per 100,000 U.S. athletes die in that manner."

SCD is the leading cause of death and death during exercise of NCAA student-athletes, with an incidence of 1 in 43,000 student-athletes per year (*Harmon et al., 2011*). HCM, involved in 36% of cases, is the leading cause of SCD of U.S. athletes (*Maron et al., 2009*). Sex and racial disparities have been reported by *Harmon et al. (2011)*, noting SCD among male student-athletes was 2.3× more common than among female student-athletes and further noting the incidence of SCD in Black student-athletes was 3× that of White student-athletes. Moreover, SCD rates vary by sport, with SCD most common in basketball, football, swimming, lacrosse, and cross-country. Prevalence of risk factors for SCD has been reported as roughly 3 in 1000 (see *Harmon et al., 2011*).

Tragic sudden cardiac deaths of youth athletes—such as 16 year old Michigan high school basketball player Wes Leonard (e.g., *Moisse, 2011*), 17 year old Colorado high school rugby player Matthew Hammerdorfer (e.g., *Sandell & Dolak, 2011*), 16 year old California high school swimmer Justin Carr (e.g., *Sondheimer, 2013*), and 23 year old Pennsylvania runner Kyle Johnson (e.g., *SCA Foundation, 2013*; *Zimmerman, 2013*)—grab local and national attention and often prompt calls for stronger protections of youth athletes. Pre-participation screening of athletes has been criticized for being too weak. Lisa Salberg, CEO of the Hypertrophic Cardiomyopathy Association (HCMA), has been a vocal critic of current screening practices stating, "As parents, we encourage our children to participate in athletics and organized sports, however are we doing enough to ensure that they are healthy enough to be playing?" (*Hirschhorn, 2010*). Other criticisms focus on the response to such sudden cardiac arrest emergencies when they do occur: public health advocates call for increased availability of automatic external defibrillators at sporting venues and highlight "success" stories such as that of Davis Nwankwo, a Vanderbilt basketball player who collapsed on the court during practice in 2006 but whose life was saved by his athletic trainer armed with an automatic external defibrillator (e.g., *Sayre, 2007*). The Sudden Cardiac Death of the Young Surveillance and Prevention Project of the Michigan

Department of Community Health Genomics Program (see *MDCH, 2012*) has been productive in studying the problem from many angles and has identified 21 action steps in five areas of need to prevent SCD, including improvement of pre-participation sports screening, training in the use of AEDs and performance of CPR, education/awareness of SCD, and both emergency response and medical examiner protocols (*Duquette & Anderson, 2009*).

Pre-participation educational/awareness and screening programs may provide important opportunities to get the public's attention on heart health issues. Cascade genetic screening (i.e., a process wherein first degree relatives are screened after a mutation has been identified in a proband) has been touted for its ability to identify carriers who may be asymptomatic or pre-symptomatic and who may benefit from early implementation of surveillance and therapeutic efforts (such as use of implantable defibrillators) (e.g., *Ackerman et al., 2011*). However, it is important to note the sports implications of identifying genetic variants in asymptomatic or pre-symptomatic individuals who currently have or may in the future have interests in competitive sports. Is identifying such genetic variants a means to ensure player safety or simply a means of facilitating systematic genetic discrimination? While expanding access to genetic risk information is critically important for prevention of SCD, numerous legal and ethical challenges lurk in the ways in which genetic risk information flows among the various actors in the sports context (e.g., from an individual athlete to the athlete's parents, health care providers, trainers, coaches, administrators, player's agents, team's scouts, league officials, media, spectators, boosters, etc.) as well as the way in which decisions are made relying upon that genetic information (ranging from absolute assumption of risks versus involuntary disqualification from participation). Numerous influences suggest that the individual athlete will *not* be in control of the decision to engage in competitive sports once one or more genetic risk variants is/are discovered — regardless of whether that discovery occurs during participation in genomic medicine and research completely unrelated to sports or during pre-participation screening specifically for sports.

For example, Pennsylvania recently adopted the *Sudden Cardiac Arrest Prevention Act* (24 P.S. §5331 *et seq.*, P.L. 574, No. 59 §1, effective July 30, 2012). This statute provides that a student-athlete must be removed from play and prevented from returning to play if the individual "is known to have exhibited *signs or symptoms* of sudden cardiac arrest *at any time* prior to or following an athletic activity..." (24 P.S. §5333(c)(2)). The statute prohibits the individual's return to athletic activity "until the student is evaluated and cleared for return to participation in writing by a licensed physician, certified nurse practitioner, or cardiologist" (24 P.S. §5333(c)(3)). While genetic counselors are notably omitted from this list, the statute permits those authorized to provide return-to-play clearances to consult other "licensed or certified medical professionals", which would include genetic counselors. This is because Pennsylvania is among the states that have passed bills requiring genetic counselors to be licensed medical professionals (63 P.S. §422.13d, P.L. 576, No. 125 §2, effective February 21, 2012; see also *Wagner, 2012*). The Sudden Cardiac Arrest Prevention Act does not define "signs or symptoms", and ordinary

parsing of the statute's text indicates "signs" has different meaning from "symptoms." Otherwise, the terms "signs" and "symptoms" are redundant and "or" is meaningless. Thus, it is possible (if not likely) that those who must comply with the Sudden Cardiac Arrest Prevention Act (i.e., public school officials, their coaches, and those medical practitioners making clearance determinations) will interpret the statute in such a way that the presence of a genetic variant—regardless of the presence or absence of symptoms expressed by the individual—would constitute a "sign" that precludes the individual from participation. A survey of the legislative activity, shown in Table 4, revealed at least 10 states that are presently considering similar legislation to Pennsylvania's Sudden Cardiac Arrest Prevention Act.

Moreover, the American College of Cardiology (ACC) Bethesda Conference #36 provide specific recommendations regarding eligibility and disqualification of competitive athletes (*Maron & Zipes, 2005*; see also *Pelliccia, Zipes & Maron, 2008*; *Maron et al., 2009*). Bethesda Conference #36 Guidelines include extensive discussion on HCM, LQTS, MFS and related cardiac conditions. Relevant guidelines are shown in Table 5. These guidelines generally recommend restricting individuals from participation in competitive sports. It is uncertain how the Bethesda Conference #36 Guidelines are intended by the ACC to be applied in cases where the diagnosis of the HCM, LQTS, MFS or related conditions are made using a genomic medicine approach available today. While the Bethesda Conference #36 Guidelines note that, at the time the guidelines were drafted, genotype alone did not warrant disqualification from competitive sports, the subsequent biomedical literature and the 2013 ACMG recommendations on reporting incidental findings may contribute to medical professionals making cautious (perhaps overly cautious or paternalistic) medical clearance decisions. Empirical data regarding how medical professionals are currently making sports clearance decisions in light of genomic medicine and research advances and recent legislative and policy activities are not yet available.

The purpose of pre-participation screening is "to provide potential participants with a determination of medical eligibility for competitive sports that is based on evaluations intended to identify (or raise suspicion of) clinically relevant, preexisting abnormalities" (*Maron et al., 2007*). Screening is for "the identification of at-risk athletes and the prophylactic prevention of cardiac events during sports by selective disqualification" (*Maron et al., 2007*). Medical professionals providing eligibility certification and making determinations of disqualification are expected to follow the American Heart Association (AHA) recommendations for student-athletes (at both high school and collegiate levels), and failure to comply may expose the medical professional to malpractice liability for an athlete's death or injury caused by an abnormality that would have been discovered had the guidelines been followed (*Maron et al., 2007*). While the AHA recommendations require pre-participation screening to consider personal history, family history, and physical examination (see Table in *Maron et al., 2007*), they have not required 12-lead electrocardiograms (ECG) or genetic screening (such as those panels offered by GeneDx for HCM, LQTS, and MFS and related conditions). Genetic screening for HCM was considered but rejected due to its cost, genetic heterogeneity, and the anticipated

**Table 4 State Survey of Legislative Activity to Prevent SCD of athletes.** Westlaw Next was used on June 19, 2013 to search for proposed and enacted legislation in the US related to sudden cardiac death prevention and athletic activity. The search was limited to the last 12 months of activity. The precise search terms used may have failed to uncover all of the legislative activity. OpenStates.org was used to verify the relevant subject matter contained in the bills located using Westlaw Next. Results specific to placement of automatic external defibrillators were not reported in this table. See also "Sudden Cardiac Arrest Legislation by State." Available at http://www.simonsfund.org/sudden-cardiac-arrest-legislation-by-state/ Last accessed June 19, 2013.

| State | Proposed? | Enacted? | State | Proposed? | Enacted? |
|---|---|---|---|---|---|
| Alabama | | | Montana | | |
| Alaska | | | Nebraska | | |
| Arizona | | | Nevada | | |
| Arkansas | | | New Hampshire | | |
| California | | | New Jersey | 2012 NJ SB 2367, intro. 12/17/12 | |
| Colorado | | | New Mexico | | |
| Connecticut | | | New York | 2013 NY SB 80, intro. 1/9/13 | |
| Delaware | 2013 DE SB 108, intro. 6/5/13 | | North Carolina | | |
| Florida | | | North Dakota | | |
| Georgia | | | Ohio | 2013 OH HB 180, intro. 5/28/13 | |
| Hawaii | | | Oklahoma | 2013 OK SB 39, intro. 2/4/13 | |
| Idaho | | | Oregon | | |
| Illinois | 2013 SB 1274, intro. 1/31/13; 2013 HB 15, intro. 12/10/13 | | Pennsylvania | | 24 PS §5331 *et seq.* (2012) |
| Indiana | 2013 IN HB 1178, intro. 1/10/13 | | Rhode Island | | |
| Iowa | | | South Carolina | | |
| Kansas | | | South Dakota | | |
| Kentucky | | | Tennessee | | |
| Louisiana | | | Texas | 2013 TX SB 379, intro. 2/5/13 | |
| Maine | | | Utah | | |
| Maryland | | | Vermont | | |
| Massachusetts | 2013 MA SB 1027, intro. 1/18/13 | | Virginia | | |
| Michigan | 2013 MI 4273, intro. 2/19/13 | | Washington | | |
| Minnesota | | | West Virginia | | |
| Mississippi | | | Wisconsin | | |
| Missouri | | | Wyoming | | |
| Subtotal | 5 | 0 | Subtotal | 5 | 1 |
| | | | Total | 10 | 1 |

**Table 5 Excerpts of the Bethesda Conference #36 Guidelines.**

| | |
|---|---|
| HCM | "1. Athletes with a probable or unequivocal clinical diagnosis of HCM *should be excluded from most competitive sports*, with the possible exception of those of low intensity (class IA). This recommendation is independent of age, gender, and phenotypic appearance, and *does not differ for those athletes with or without symptoms*, LV outflow obstruction, or prior treatment with drugs or major interventions with surgery, alcohol septal ablation, pacemaker, or implantable defibrillator." (p. 1341, emphasis added)<br>"2. Although the clinical significance and natural history of genotype positive-phenotype negative individuals remains unresolved, no compelling data are available at present with which to preclude these patients from competitive sports, particularly in the absence of cardiac symptoms or a family history of sudden death." (p. 1341) |
| LQTS | "2. Asymptomatic patients with baseline QT prolongation (QTc of 470 ms or more in males, 480 ms or more in females) should be restricted to class IA sports. The restriction limiting participation to class IA activities may be liberalized for the asymptomatic patient with genetically proven type 3 LQTS (LQT3)." (p.1362)<br>"3. Patients with genotype-positive/phenotype-negative LQTS (i.e., identification of a LQTS-associated mutation in an asymptomatic individual with a nondiagnostic QTc) may be allowed to participate in competitive sports. Although the risk of sudden cardiac death is not zero in such individuals, there is no compelling data available to justify precluding these individuals (who are being identified with increasing frequency) from competitive activities. Because of the strong association between swimming and LQT1, *persons with genotype positive/phenotype-negative LQT1 should refrain from competitive swimming.*" (p. 1362, emphasis added) |
| MFS and related conditions | "1. Athletes with Marfan syndrome can participate in low and moderate static/low dynamic competitive sports (classes IA and IIA) if they do not have one or more of the following:<br>a. aortic root dilatation . . .<br>b. moderate-to-severe mitral regurgitation<br>c. family history of dissection or sudden death in a Marfan relative. . ." (p. 1342)<br>"3. *Athletes with Marfan syndrome*, familial aortic aneurysm or dissection, or congenital bicuspid aortic valve with any degree of ascending aortic enlargement. . .also *should not participate in sports that involve the potential for bodily collision.*" (p. 1342, emphasis added)<br>1. Athletes with mild or moderate AR [aortic regurgitation], but with LV end-diastolic size that is normal or only mildly increased, consistent with that which may result solely from athletic training, can participate in all competitive sports. . . . Those with asymptomatic nonsustained ventricular tachycardia at rest or with exertion should participate in low-intensity competitive sports only (class IA). . ." (p. 1337) |

frequency of false-negative results (*Maron et al., 2007*). Given the nuances of the current biomedical understanding of various cardiac conditions (including HCM, LQTS, MFS and related conditions), genetic risk information is *informative* but far from *determinative* of a player's health and corresponding risks of SCD. Accordingly, it seems unwise to base decisions of athletic eligibility (particularly eligibility of those individuals who are asymptomatic or pre-symptomatic) on the presence of a genetic variant alone – even if such genetic variants are covered by the ACMG's recommended list for reporting of incidental findings.

### Facilitating player safety or genetic discrimination?

Together, the ACMG recommendations for reporting incidental findings, the Sudden Cardiac Arrest Prevention Act in Pennsylvania (and similar state bills if adopted), the Bethesda Conference #36 Guidelines, and general aversions to tort liability risks (e.g., medical malpractice) will work together in a conservative, paternalistic manner. Undoubtedly such efforts may promote *individual* safety; however, it would be inaccurate to characterize these efforts as promoting *player* safety. Ultimately the majority of individuals "protected" will not be permitted to be players at all, since adequate medical clearance paperwork will be

hard to come by. Pre-participation screening practices that serve to disqualify individuals from participation in sports as a result of family medical history or genetic information when they themselves are asymptomatic is genetic discrimination (specifically, systematic disparate treatment).

Notably, racial disparities exist in sports contexts, including disparate participation in specific sports and in specific positions in team sports (e.g., *Graves, 2005*). Racial disparities exist among athletes as well as among executives/decision-makers. For example, racial minorities represent more than two-thirds (69%) of all NFL players but hold few positions of authority, with racial minorities representing 0% of CEOs/presidents; 3% of majority ownership; <20% of head coaches or general managers; and <15% of physicians and head trainers (*Lapchick et al., 2012*). The disproportionate participation may be reversed in other sports (e.g., skiing and snowboarding (e.g., *Eiss, 2011*)). Many factors contribute to existing disparities in sports (e.g., sociocultural factors of differential power, wealth, and prestige given to sports; self-selection biases; economic factors influencing access to educational, nutritional, and training resources; environmental factors including in utero exposures; and genetic factors). The existing disparities are an important aspect of the context in which genetic/omic technologies are being integrated in sports training and medicine programs. Understanding this context is essential when evaluating potential challenges arising from this integration of genetic information in sports and the development of "genomic sports medicine." The genomic revolution has the potential to alleviate or exacerbate racial disparities depending upon how the integration is executed. Enhancing the awareness and understanding of genetic risks and increasing access to health information through pre-participation screening for sports is a means to reach under-served minority populations. However, that health opportunity could translate into employment discrimination, quashing future professional sports employment opportunities via disproportionate selective disqualifications and, in the process, create life-changing psychological, social, professional, and economical impacts for members of those under-served minority populations.

## Sports-related genetic/omic screening

The integration of genetic/omic technologies in sports contexts is not a hypothetical concept. Various renditions have already been attempted both domestically and internationally for diverse purposes (*Wagner, 2013*). Lawsuits can be strong motivators for the adoption of DNA screening policies. Such efforts promote safety by identifying risks prior to participation, but screening efforts also serve to limit liability exposure of the event organizers, the leagues, the teams, the schools, etc. Such was the case for the NCAA's adoption of sickle cell carrier screening. The NCAA sickle cell screening policy arose as part of a settlement agreement resolving a lawsuit that followed the death of Dale Lloyd II, a 19 year old football player at Rice University. The student-athlete died from acute exertional rhabdomyolysis (a complication of sickle cell trait) after collapsing on the field upon running 100-yard sprints (e.g., *Zarda, 2010*). The NCAA implemented the screening policy notwithstanding criticism and opposition from the American Society of

Hematology (ASH), as the ASH favors universally applicable safety policies to help athletes regardless of sickle cell carrier status (e.g., *Petrochko, 2012*). The NCAA subsequently expanded its screening program beyond Division I athletes to include athletes in Divisions II (*Hendrickson, 2012*) and III (*Brown, 2013*).

Genetic screening may be forthcoming for concussion risks (e.g., APOE $\varepsilon4$), if the NFL adopts concussion screening as part of settlement agreements for pending lawsuits involving more than 4200 players (*In Re: National Football League Players' Concussion Injury Litigation*; see also *Anderson, 2013*) or if the NCAA reacts to the American Academy of Neurology updated guidelines naming APOE as a risk factor for chronic neurobehavioral impairment (*Giza et al., 2013*) and expands its screening program beyond sickle cell carrier status. Some have already suggested compulsory brain scans and genetic testing for boxers and others have suggested banning such sports altogether (e.g., *Jordan, 1998*; *Spriggs, 2004*; *Kelland, 2013*). At least one company that had provided sports-related personal genomics services DTC (Athleticode) has repositioned itself to focus specifically on concussion risks.

Genetic screening may also be forthcoming for HCM risks (e.g., MYH7, MYBPC3, and TNNT2) as sudden cardiac deaths of seemingly healthy youth (e.g., most recently Kyle Johnson) prompt public calls for pre-participation screening. As mentioned earlier, about one-third of athletic field deaths in the United States are caused by HCM, an abnormal thickening of the ventricular walls of the heart muscle (*Maron, 2005*). Screening for HCM has been advocated not to promote safety of those engaged in athletics but, rather, to identify and disqualify individuals prior to participation (e.g., *Corrado et al., 1998*). Distinguishing potentially life-threatening HCM from "athlete's heart", a non-pathological condition resulting from intense training, is quite challenging (e.g., *Maron, 2005*; *Cheng, 2009*; *Creswell, 2009*). Despite potential false negatives, genetic testing is the "most definitive way" to distinguish HCM from athlete's heart (*Cheng, 2009*). While more than 1000 distinct mutations have been identified, the mutations found within two genes (MYH7 and MYBPC3, OMIM #160760 and #600958, respectively) collectively account for more than 75–80% of HCM (*Ho, 2011*). In 2010 a jury granted an award of $1.6 million dollars to the parents of Antwoine Key, a 22 year old basketball player for Eastern Connecticut State University who collapsed and died as a result of undiagnosed HCM. The student-athlete had been given clearance to play by five different physicians notwithstanding a heart murmur noted during a pre-participation exam more than three years prior to his collapse (*Key v. Abdulah, 2010*; see also *Morlan, 2010*; *Hirschhorn, 2010*).

## Moving toward athlete-initiated integrations of personal genomics in sports

Personal genomics (PGx) services of varying scope and quality (in terms of analysis, return of results, and interpretations) are now available direct-to-consumers. These PGx services include and sometimes focus on traits and conditions relevant to sports (*Wagner, 2013*; *Wagner & Royal, 2012*; JK Wagner and CD Royal, unpublished data), allowing individual athletes to access information (a) without coordination with a school, team, or perhaps

even parents and (b) specifically outside of the formal and potentially intimidating context of medical care, scientific research, and pre-participation screening processes. Among the ∼250 genetic variants implicated in sports-related phenotypes (*Rankinen et al., 2010*; *Bray et al., 2009*; *Rankinen et al., 2006*; *Wolfarth et al., 2005*; *Rankinen et al., 2004*; *Pérusse et al., 2003*; *Rankinen et al., 2002*; *Rankinen et al., 2001*) are *ACTN3, COL5A1, COL12A1, COL1Al, GDF5, ACE, ADRB2, PPARGC1A, MMP3, APOE, MYH7, MYBPC3, TNNT2, DIO1, NOS3, IL6, VEGFR, HIF1, MCT1, EPOR*, and *SCN5*, all of which have been available as part of one or more DTC sports-related genetic tests (*Wagner & Royal, 2012*). Many of these genetic variants have been characterized as "gene doping targets" (*Azzazy, Mansour & Christenson, 2009*) or "candidate genes for sport doping" that will enable "the creation of a superman or superwoman athlete" through "well-placed genetic physiologic tweaks" (*Gaffney & Parisotto, 2007*). However, anthropological geneticists have cautioned that genotype is not the full explanation of "what makes a champion", as there are many potential confounders, including spurious associations due to population structure or ancestry and the complex and varying effects of gene-environment interactions over an individual's lifetime (*Brutsaert & Parra, 2006*).

It is evident that the sports-related sector of the larger PGx industry has considerable potential for diversity in terms of phenotypes of interest and genetic loci assayed. Previous research identified the companies in the sports-related PGx industry and analyzed information provided online to prospective consumers (*Wagner & Royal, 2012*). An empiric investigation involving participant observation and the purchase of four sports-related DNA tests in 2011 (JK Wagner and CD Royal, unpublished data) revealed that there is, in fact, considerable diversity in offerings and that there simply is no "typical" or "standard" sports-related DTC panel for the PGx industry (see Table 6). PGx offerings for genetic information related to power or strength have focused on *ACTN3* (rs1815739, OMIM #102574), *IL6* (rs1800795, OMIM #147620), and *DIO1* (rs11206244 and rs12095080, OMIM #147892). For endurance, PG offerings have focused on *NOS3* (rs2070744 and rs1799983, OMIM #163729), *HIF1A* (rs11549465, OMIM #603348), *MCT1* (rs1049434, OMIM #600682), *VGFR* (OMIM #191306), and *EPOR*(rs121918116, OMIM #133171). PGx offerings for soft tissue injury risks have focused on *MMP3* (rs679620, rs591058, rs650108, OMIM #185250), *COL5A1*(rs12722, OMIM #120215), *COL12A1* rs970547, OMIM #120320), *COL1A1* (rs1800012, OMIM #120150), and *GDF5* (rs143383, OMIM #601146). For cardiac risks, PGx offerings have focused on MYH7 (OMIM #160760), MYBPC3 (OMIM #600958), and SCN5A (rs7626962, OMIM #600163). PGx offerings have even included concussion risks, focusing on APOE (OMIM #107741). Upon closer examination, it is apparent that AIBiotech has provided access to a number of genes on the ACMG's recommended incidental findings report: AIBiotech has provided exome sequencing of *MYH7, MYBPC3*, and *TNNT2* (three of the genes related to HCM risks) and testing of *SCN5A* (a gene on the ACMG's list of recommended reporting for LQTS and related acquired arrhythmias). *SCN5A*, a cardiac sodium channel gene with a number of common polymorphisms (OMIM #600163) (*Cheng et al., 2011*), contains a Y1002 variant (S1103Y, rs7626962) that is rare in European Americans (*Smith et al., 2011*) but reported at

**Table 6 Summary of Genes Analyzed by Four DTC sports-related tests in 2011.** Direct participant observation occurred in May 2011. The author purchased the four tests from US-based companies identified previously (*Wagner & Royal, 2012*) and submitted her own DNA for analysis. Data on the following variables of interest were collected as part of the participant observation: the purchase process (e.g., informed consent requirements, details regarding terms of service and privacy policies); the DNA collection process (appearance of packaging; educational or marketing literature included; type of specimen required; supplies and instructions); timing issues (estimated wait times; lag times between dates of purchase, kit receipt, DNA sample arrival at company, and results receipt); and issues related to representation and return of results (e.g., media format used; type of information reported; manner in which risk scores or performance predictions were represented; descriptions of methods; readability of results material; and availability of interpretation support). Notably, the motivations, reactions, perceptions of satisfaction, comprehension of results, and other interesting facets of the consumer experience were not the focus of that study: rather, the research was conducted to permit a data-driven discussion of DTC sports-related genetic testing aspects that occur after customers click "purchase." The information reported in Table 6 is a summary of those specific tests as provided in May 2011 using the author's DNA. Tests may have changed. For example, as of late 2012, Athleticode was no longer offering this particular test.

| "SportsXFactor" by AIBiotech | "Body Scope Kit" by Athleticode | "Athletic Profile Test" by Warrior Roots | "Atlas First" by Atlas Sports Genetics |
|---|---|---|---|
| DIO1 | COL1A1 | ACTN3 | ACTN3 |
| NOS3 | COL51 | HIF1 | |
| IL6 | COL12A1 | MCT1 | |
| ACTN3 | GDF5 | ADRB2 | |
| PPARGC | MMP3 | DIO1-D1A | |
| VEGFR | | DIO1-DIB | |
| COL1A1 | | NOS3 | |
| COL12A1 | | PPARGC1A | |
| COL5A1 | | ACE | |
| MMP3 | | EPOR | |
| APOE | | | |
| MYH7 | | | |
| MYBPC3 | | | |
| TNNT2 | | | |
| SCN5A | | | |

allele frequencies ∼13.2% in African Americans (*Splawski et al., 2002*). This polymorphism reportedly causes a "small but inherent and chronic risk of acquired arrhythmia" (*Splawski et al., 2002*).

While 23andMe is not focused on providing a sports-related service, its PGx analysis does incorporate much of the information athletes could seek elsewhere. 23andMe provides reports on muscle performance, response to diet, response to exercise, pain sensitivity, asthma, sickle cell carrier status, heart rhythm disorders, and APOE4 status (though not reported for concussion risk (see *Bethann, 2013*). The 23andMe's latest platform (version 3) has considerable coverage of the recommended ACMG list shown previously in Table 3. Browsing raw data in 23andMe for the first three genes of each condition listed in Table 3, the PGx service includes 33 SNPs in *MYBPC3*, 52 SNPs in *MYH7*, and 30 SNPs in *TNNT2* (three genes on ACMG's list for HCM); 271 SNPs in

*KCNQ1*, 50 SNPs in *KCNH2*, and 116 SNPs in *SCN5A* (three genes on ACMG's list for LQTS and related arrhythmias); and 139 SNPs in *FBN1*, 26 SNPs in *TGFBR1*, and 75 SNPs in *TGFBR2* (three genes on ACMG's list for MFS and related disorders). The consumer's or user's ability to download the raw data from 23andMe for subsequent independent analysis and interpretation makes it attractive for individuals whose motivations may include sports-related purposes.

Collegiate sports medicine and athletic programs are beginning to incorporate PGx as well. One example is Dr. Stuart Kim's work to integrate PGx for injury risk prevention at Stanford University (see https://sportsgenetics.stanford.edu/). The Stanford pilot program focuses on secondary analysis and interpretation of 23andMe raw data files (version 3 platform) to assess risks for stress fractures, ACL ruptures, Achilles tendinopathy, disc degeneration, hemoglobin count, vitamin and mineral deficiencies, and osteoarthritis as well as sickle cell carrier status. Participants upload their 23andMe raw data to the program's website and Dr. Kim and his colleagues perform secondary analysis of the data. The program involves one-hour consultations to discuss the results, and the idea is that athletes could modify their training programs, take nutritional supplements (e.g., Vitamin D), and obtain additional medical services to prevent disruptions to their athletic seasons due to injury. At least 15 elite athletes from the Stanford Triathlon Club have participated thus far. This pilot program is expected to expand to include the Varsity cross-country teams at Stanford and the University of California Los Angeles before being proposed for adoption by the Pac-12 Conference at large (S Kim, personal communication; March 15, 2013) and the program has also initiated recruitment of endurance athletes in the San Francisco Bay Area.

## Recognizing the vulnerabilities of student-athletes

Individuals with extraordinary athletic abilities are rarely thought of or described as "vulnerable" (but see *Davis, 1998*). Nonetheless, when considering the potential and probable applications of PGx in sports contexts, a number of characteristics suggest the vulnerability (see, e.g., *Beauchamp & Childress, 2009*) of student-athletes. To ensure (1) respect for autonomy of the student-athletes, (2) optimization of the risks and benefits, and (3) promotion of justice among student-athletes, recognition of these vulnerabilities suggests heightened care be exercised when developing a program incorporating PGx technologies. These vulnerabilities of student-athletes highlight potential reasons why a DTC market will and should continue to exist (and why best practices should be promoted within the broader PGx industry), as individuals interested in sports should have a means to access their own PGx information without other actors (i.e., parents, coaches, league officials, medical providers, etc.) automatically and/or simultaneously gaining unfettered access to those data.

First, student-athletes are often under the age of majority (i.e., 18 or 21 years of age), which effectively limits the individuals' ability to enter into contracts or provide consent independently and thereby perpetuates a state of dependency upon parents or guardians. Regulatory protections for participation in research include the requirement

that researchers obtain not only informed assent from minors but corresponding informed consent from the minors' parents or guardians. Considerable deference is given to parents in the U.S. to make decisions in the "best interest of the child" (see, e.g., *Troxel v. Granville*). Nonetheless, parents may hold extreme perspectives regarding PGx and/or sports that could impinge on the student-athletes' ability to effectuate their needs and interests in this context. For example, parents espousing genetic determinism could alternatively deny or compel their children to obtain PGx information for sports purposes. Economic dependency of student-athletes further exacerbates the potential compromise of autonomous decision-making by student-athletes who are considering the use of PGx in sports contexts.

Coercive pressures to participate in any activity perceived to provide a competitive edge along with the intensely competitive personalities of student-athletes create vulnerabilities for student-athletes as well. While coercive pressures of the business of sports should be acknowledged when present, they cannot be assumed *a priori* as undue influences that negate or preclude voluntary consent from student-athletes. Red flags raised by existing coercive pressures, however, suggest that recruitment procedures include education and outreach efforts to explain clearly the details of the program and further suggest that a reasonable waiting period be provided to ensure that decisions to participate are made deliberately and upon disclosure of adequate information.

A third characteristic of vulnerability is the precarious status that student-athletes hold under the Genetic Information Nondiscrimination Act (GINA). Title II of GINA prohibits employers and labor organizations from requesting or using genetic information in employment decisions. As was previously examined elsewhere, sports employment is not exempted from GINA Title II's prohibition on using genetic information to limit, classify, segregate employees (i.e., players), and employers enjoy no defense for use of genetic information as a "bona fide occupational qualification" (*Wagner, 2013*). GINA coverage, however, does not clearly situate student-athletes under the umbrella of a protected class. Rather, GINA protections extend only to former employees, current employees, and prospective employees. Current amateur rules and legal fictions combine forces in ways that undermine the general labor and genetic nondiscrimination protections of student-athletes. In exchange for the opportunity for a position on the team, the privilege to wear the uniform, and the benefits of education and scholarships, student-athletes transfer any monetary benefits of their hard-earned labors to others (i.e., they waive earnings) and must refrain from gainful employment elsewhere (i.e., they agree to broad-sweeping non-compete covenants). Notwithstanding this exchange of ongoing benefits and obligations, student-athletes are denied status as school "employees" (see, e.g., *Maisel, 2011*; *McCormick & McCormick, 2006*; see also *Branch, 2011*). While numerous arguments can be made to extend GINA status as "current employees" (with schools as the covered entities) and as "prospective employees" (with professional teams as the covered entities) to student-athletes for purposes of GINA, this uncertain status exposes student-athletes to genetic discrimination as a foreseeable consequence (a) of participation

(or non-participation) in sports programs integrating PGx (such as the Stanford program) and (b) of the application of genetic technologies in pre-participation screening programs.

Finally, student-athletes are often treated with celebrity status (locally, regionally, nationally, and internationally), creating privacy concerns. Privacy rights in the U.S. are protected contextually (not with one broad-sweeping statute), and are balanced with other interests (such as First Amendment freedoms of speech and association). While one need not "live the life of a recluse" to enjoy privacy rights (*Maxey, 1937*), celebrity status broadens that which is newsworthy and is sometimes considered a waiver of some privacy rights (e.g., *Hull v. Curtis Publishing Co*). The integration of genetic screening programs or use of PGx in sports could undermine the individual's genetic privacy interests and expose the individual to a number of future risks, including discrimination (see, e.g., *Quick, 2012*).

## CONCLUSIONS

While some may claim the limited scope of the ACMG recommendations (i.e., WGS/WES in clinical settings) indicates that very few individuals would fit the problematic scenario outlined here, that assumption is not supported by any data. Moreover, that assumption fails to consider that those performing WGS/WES in non-clinical settings may, too, look to the ACMG's recommendations for guidance on what the professional standard of conduct should be. The possibility of discovering "incidental findings" in the ACMG's list of genes with WGS/WES of normal, unaffected, or asymptomatic individuals is not negligible. Healthy individuals, as a result, might be precluded from participation in sports unfairly as a result of the conservative influences of the Sudden Cardiac Arrest Prevention Act in Pennsylvania (and similar state statutes if adopted), the Bethesda Conference #36 Guidelines, and the broader medical profession's aversion to tort liability risks. Even when medical clearances to play are provided by medical professionals, genetic illiterate or genetic deterministic parents may serve as distinct barriers for sports participation. Updated guidelines clarifying how the ACMG's recommended list of reporting incidental findings should be reconciled with these conservative forces causing medical professionals to err on the side of caution in sports contexts (i.e., disqualification or restriction from athletic activity). It is possible that surveillance—not disqualification or restriction from participation in competitive sports—is an appropriate measure for asymptomatic individuals who happen to carry a variant in one of the genes noted in Table 3. Such an approach acknowledges the public health interests in preventing sudden cardiac death but balances those interests in a way that respects the individuals' personal interests in assuming the risks and playing competitive sports notwithstanding the presence of a genetic risk variant. Individuals will vary in their reactions to learning personal genetic information. Within reason, those individual reactions and preferences for risk assumption should be respected.

When integrating genetic technologies into sports settings, it is absolutely essential that the intended purposes and unintended consequences are carefully considered during program development and that these are subsequently re-evaluated on a recurring basis throughout implementation. While official guidelines for genomic athletic programs or

genomic sports medicine programs are not yet available, a few preliminary suggestions may be articulated here. First, the goals and purposes of the integration of PGx into any sports medicine or athletic program must be clearly defined. Identifying the distinct motivations for and interests in PGx in any sports system (such as the individual athletes seeking empowerment through more personalized pre- and re-habilitation strategies, the coaching personnel seeking information with which to consider talent and injury risks when distributing limited athletic scholarships, and the trainers and medical personnel seeking improved care strategies) is key to preparing the system to avoid unintended consequences and facilitate achievement of its intended purposes. Any integration at the collegiate level will have downstream effects if/when athletes attempt to transition to the professional sports level. Care must be exercised to avoid disparate treatment and impacts that could exacerbate existing inequities (including racial disparities). Second, transparency of the authorized flow of information is integral to a successful program. The extent to which PGx information is accessible and/or shared within the personnel structure of a program affects student-athletes' rights and interests. Prior to any implementation of PGx in sports medicine and athletic programs, designers must make deliberate decisions about the intended direction and volume of the informational flow for the system. Transparency promotes not only compliance with HIPAA/HITECH data security and privacy requirements but also respect for contextual integrity. Third, "terms of use" must be established for all potential decision-makers in the program. Distinct categories of use should be clearly articulated, such as (1) required, (2) recommended, (3) authorized/permissive, and (4) prohibited uses. Terms of use should reflect specific roles of trainers, coaches, and administrators within such programs and also specific contexts in which PGx information might be valuable. Enforcement mechanisms should be articulated. These enforcement mechanisms could incorporate disciplinary standards (such as loss of authorization/access) and could incorporate provisions for liquidated damages for the individual victims of the violations (carefully structured such that any payout avoids jeopardizing the athlete's "amateur" status). Fourth, such programs should adopt policies and procedures to minimize opportunities for the voluntariness of a decision to participate to become compromised. The existence of coercive pressures alone (from peers, parents, coaches, etc.) does not preclude voluntary participation in a PGx sports medicine or athletic program; however, such pressures do present challenges for obtaining voluntary assent/consent. Education of athletes, trainers, coaches, administrators, and media relations for each program could alleviate challenges by establishing reasonable expectations and behavioral norms. Finally, program administrators should engage players' associations and advocates for student-athletes. Each stage (design, implementation, and operations) of programs that integrate PGx would benefit from periodic review by advocates for the athletes and from regular feedback from players' associations regarding issues emerging with athletes' transition to the professional sports level.

While promoting player safety is a laudable goal, facilitating unfettered genetic discrimination of asymptomatic athletes is an unfortunate but likely outcome under

present policy conditions. Given disparate rates of participation of individuals of racial minorities in particular mainstream sports (e.g., basketball, football, baseball, soccer, etc.), there is a valuable opportunity to provide access to education and genetic risk information (about sudden cardiac death and other serious medical conditions) to individuals and their families (and by extension their communities). This opportunity also suggests there is ample room for disparate discriminatory impacts if decisions based on that screening information (e.g., disqualification from participation) are mandated in fact or in practice. Moreover, the adoption of "fair competition" policies is precisely to restrict normal variation in athletic abilities. The co-existence of the Olympic Games, Paralympics, and Special Olympic Games is an obvious example, though sex segregation, age levels, and weight classes are demonstrative as well. Policies to disqualify, limit, or otherwise classify athletes on the basis of carrying a genetic risk variant is essentially the beginning of genetic segregation in sports. Such policies—even when attempting to serve public health goals—reify public notions of genetic determinism. Moreover, because the distribution of genetic diversity today is a function of our demographic and evolutionary past, such policies could disproportionately distribute risks and benefits of the integration of PGx in sports.

Finally, the integration of PGx in sports must be considered as part of a broader data-rich movement. The "Information Age" has enabled increasingly personalized diet and exercise regimes, shaped by personal data generated on mobile devices that monitor nutritional intake, activity levels, and sleep patterns and subsequently shared online with family, friends, teammates, and strangers. While serious athletes are likely early-adopters of PGx (as high rewards may be gained from small improvements in performance), casual athletes may be quick to follow. The public's reception of PGx in sports medicine and athletic programs will be dependent upon genetic literacy and will be influenced (positively or negatively) by the program's design (e.g., the care with which biopsychosocial, legal, and ethical challenges are anticipated and successfully managed). Waiting to educate individuals about genetic information until they are in a clinic as patients (or parents of patients), in a laboratory as research participants, or in an attorney's office or administrative setting facing specific, life-altering decisions (from end-of-life care in light of APOE4 Alzheimer disease risks to participation in competitive sports in light of sudden cardiac arrest risks) is simply too late. Genomic sports medicine and PGx in athletic settings provide an extraordinary opportunity to improve awareness of sudden cardiac death and improve genetic literacy generally. The Ad Council has provided important and effective public service announcements for generations (e.g., the Smokey the Bear campaign in 1944 helping kids prevent forest fires; "A mind is a terrible thing to waste" campaign supporting the United Negro College Fund in 1972; McGruff the Crime Dog taking a bite out of crime in 1978; and the Crash-Test dummies encouraging kids to buckle safety-belts in 1985) (*Ad Council, 2013a*). In 2005 the Ad Council launched a Coalition for Healthy Children to fight childhood obesity using campaigns geared toward kids and parents, including "Be a Player: Get up and Play an Hour a Day" and "Eat well. Play hard. Make it balance" campaigns (*Ad Council, 2013b*). Similarly, numerous

league-affiliated initiatives (e.g., NFL's "Play 60" or "NBA Cares") are designed to "give back" to communities and improve health, fitness, and quality of life of kids. Pilot campaigns and empiric data are needed to explore whether great strides could be made in improving genetic literacy via exposure to the topic via the public's enthusiastic interest in sports. Whether access to PGx information will empower athletes or oppress them is not a predetermined outcome. To enable the former and prevent the latter requires active policy development rather than passive observation.

## ACKNOWLEDGEMENTS

The author is grateful to Reed Pyeritz for his generous mentoring support and to Shana Merrill, Debra Duquette, Jessica Mozersky, and Heather Norton for constructive dialogue on this research topic.

### Funding

Preparation of this article was supported in part by Grant No. K99HG006446-02 and by Grant No. P50HG004487-05 from the National Human Genome Research Institute (NHGRI). Purchase of the four underlying sports-related DNA tests was possible in 2011 thanks to funding support from Dr. Charmaine Royal at the Duke Institute for Genome Sciences & Policy at Duke University. The content of this article is solely the author's responsibility and may not represent the official views of the author's funding sources or employers. The funders had no role in study design, data collection and analysis, decision to publish, or preparation of the manuscript.

### Grant Disclosures

The following grant information was disclosed by the authors:
National Human Genome Research Institute (NHGRI): Grant Numbers K99HG006446-02, P50HG004487-05.

### Competing Interests

Jennifer K. Wagner is an Academic Editor for PeerJ and has no other competing interests to declare.

### Author Contributions

- Jennifer K. Wagner conceived and designed the experiments, performed the experiments, analyzed the data, contributed reagents/materials/analysis tools, wrote the paper.

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
