# Peer review of "Playing with heart and soul...and genomes: sports implications and applications of personal genomics"

_PeerJ, doi:10.7717/peerj.120_

## Round 0.1 · original submission · Minor Revisions

As you can see below, the reviewers are very positive about your paper, but have a number of minor suggestions for improvement. Comments 1 and 3 from Reviewer 1 are very much worth paying attention to, as is the suggestion for a careful proof-reading to get rid of typos, grammatical issues and the like. If you can successfully address the reviewer's comments, this should be ready to be accepted.

·

Basic reporting

This is an impressively comprehensive report on the use of genome and exome sequencing as it relates to athletes. Its focus is on health risks that might be uncovered by such tests rather than their use or validity for predicting athletic talent. The article draws out anthropological, ethical and legal implications for such testing.

Experimental design

As a literature review and analysis this article easily meets the standards for comprehensiveness and transparency that are fundamental to such undertakings.

Validity of the findings

I anticipate that this article will become a must-read for anyone interested in personal genomics and sport. The analysis is for the most part carefully drawn and persuasive. I would ask AU to consider the following:
1. The very interesting aggregate data on p. 3 about risks does not answer a questions this reviewer found most compelling: What is the risk to an individual with each of these genes of dying with versus without participating in athletics? Is it possible to answer this question?
2. On p. 12 AU chooses derisive language to describe the situation of NCAA athletes: "...arbitrary and oppressive rules of amateurism"; "...modern servitude of student athletes." This line of criticism is by now well established but far from universally accepted. It is worth noting that the great majority of student athletes, and the overwhelming majority of woman college athletes, are in non-revenue generating sports. The validity of AU's argument does not depend on such contentious editorializing, which is more of a distraction than a central point.
3. Despite the great detail in the article, I had some difficulty understanding how AU would deal with cases such as those in the Bethesda #36 Guidleines, which state:
"This recommendation is independent of age, gender, and phenotypic appearance, and does not differ for those athletes with or without symptoms." In general the article would benefit from clear articulation of standards for the use of personal genomic information by athletes, parents (of minors), and other parties interested in such information (coaches, university risk managers, etc.) The article has much to say about these cases, but it would be very helpful to have a clear, brief summary.

Additional comments

An admirable piece of work. The paper would benefit from one more careful proofreading to pick up the errors that dot the paper and distract from the important points it makes.

·

Basic reporting

This is very well written editorial which addresses an important issue in sports medicine/psychiatry. My only critique/suggestion would be to offer a bit more explanation of genomic sports. It is discussed, but many readers may not be familiar with the concept.
I happen to agree with her analysis of this potential issue.

Experimental design

N/A, Op ed piece

Validity of the findings

N/A, Op ed piece

Additional comments

This is a nicely written opinion paper. I fully agree with the points you made, but would offer just a bit more background information for the reader.

---

## Round 0.2 · accepted · Accept

Appreciate your responsiveness to the reviewers' comments.